# Elucidating the Response of Crop Plants towards Individual, Combined and Sequentially Occurring Abiotic Stresses

**DOI:** 10.3390/ijms22116119

**Published:** 2021-06-06

**Authors:** Khalid Anwar, Rohit Joshi, Om Parkash Dhankher, Sneh L. Singla-Pareek, Ashwani Pareek

**Affiliations:** 1Stress Physiology and Molecular Biology Laboratory, School of Life Sciences, Jawaharlal Nehru University, New Delhi 110067, India; khalidbiochemistry@gmail.com (K.A.); joshirohit6@gmail.com (R.J.); 2Division of Biotechnology, CSIR-Institute of Himalayan Bioresource Technology, Palampur 176061, India; 3Stockbridge School of Agriculture, University of Massachusetts Amherst, Amherst, MA 01003, USA; parkash@umass.edu; 4Plant Stress Biology, International Centre for Genetic Engineering and Biotechnology, New Delhi 110067, India; sneh@icgeb.res.in; 5National Agri-Food Biotechnology Institute (NABI), Mohali 140306, India

**Keywords:** abiotic stress, climate change, combined stress, drought, flooding, heat, salinity, sequential stress

## Abstract

In nature, plants are exposed to an ever-changing environment with increasing frequencies of multiple abiotic stresses. These abiotic stresses act either in combination or sequentially, thereby driving vegetation dynamics and limiting plant growth and productivity worldwide. Plants’ responses against these combined and sequential stresses clearly differ from that triggered by an individual stress. Until now, experimental studies were mainly focused on plant responses to individual stress, but have overlooked the complex stress response generated in plants against combined or sequential abiotic stresses, as well as their interaction with each other. However, recent studies have demonstrated that the combined and sequential abiotic stresses overlap with respect to the central nodes of their interacting signaling pathways, and their impact cannot be modelled by swimming in an individual extreme event. Taken together, deciphering the regulatory networks operative between various abiotic stresses in agronomically important crops will contribute towards designing strategies for the development of plants with tolerance to multiple stress combinations. This review provides a brief overview of the recent developments in the interactive effects of combined and sequentially occurring stresses on crop plants. We believe that this study may improve our understanding of the molecular and physiological mechanisms in untangling the combined stress tolerance in plants, and may also provide a promising venue for agronomists, physiologists, as well as molecular biologists.

## 1. Introduction

In the coming decades, a significant rise in agricultural productivity will be required to meet the food requirements of ~800 million undernourished people, the number which has been further growing at an alarming pace, along with the shrinking arable land [1,2,3]. In addition, under changing climatic scenarios, this challenge is further exacerbated with predicted aggravation in the frequency and magnitude of extreme and unpredictable weather events, i.e., high temperature, drought, salinity and flooding that adversely affect crop productivity and global ecosystem diversity [3,4,5,6]. Hence, to ensure global nutritional and food security, development of climate-resilient crops is the need of the hour. Most abiotic stresses, occurring either in combination or sequentially, adversely influence the earth crust by modifying the physico-biochemical properties of water, soil, atmosphere, and consequently, plants face hostile conditions [4]. Thus, crop plants continuously face various combinations or sequences of diverse abiotic stresses under field conditions [7]. Such combined or sequential stresses elicit unique acclimation responses that cannot be observed by the application of any of the stress in isolation [7]. Combined or sequential occurrences of abiotic stresses can damage the crops more significantly than their individual occurrences during various developmental stages [8,9]. In response to these abiotic stresses, plants develop innumerable physiological, biochemical, cellular and molecular mechanisms to sense and respond against different abiotic stresses [9,10]. Until now, most of the the plant-environment interaction studies have been focused on crop responses against individual stress, which usually could not be replicated in a similar manner under actual field conditions where a complex interplay of multiple stresses occur, either in combination or sequentially [11].

Field conditions are often difficult to mimic experimentally as the outcome of combined or sequential stress significantly depends upon numerous factors, including the developmental stage, stress duration, severity, and sequence of individual stresses [12,13]. Recent evidence has shownthat combined or sequential stress may affect plant metabolism differently from individual stresses, and hence, shows several unique and common responses [14]. Table 1 presents representative examples where processes associated with changes in transcriptome, proteome, metabolome and morpho-physiology of the plants in response to combined high temperature and drought (HT+D) has been studied.Moreover, various stress regulatory genes and their signaling pathways have already been classified, depending upon the interaction between individual stresses [9]. However, little information is available onthe interaction between multiple stresses in plants [15].

Plants are sessile organisms and have developed a remarkable capability to inhabit ecological niches that are regulated by their edaphic, as well as infrequent, heterogenous and chronic climatic extremes that have cyclic patterns [31]. Therefore, understanding the mechanisms underlying combined or sequential stress responses is crucial for discovering novel strategies and tools for the development of plant-resilient to suboptimal field conditions to ensure global food security and livelihood of billions of people [9,32]. Tremendous variations exist both within, and across, plant species in their ability to cope with these stresses. Technological advancements and modern genomic strategies developed over the last few decades have substantially improved our understanding of plant stress adaptation and acclimation. This has enhanced our basic understanding ofthe environmental and genetic interactions that play key roles in plant adaptation and yield stability [31].

The time has come to combine the views of plant physiologists, breeders and molecular biologists on combined or sequential stress, in order to gain a holistic understanding of the process to improve contemporary research prospects for sustainable crop ecosystems and improve productivity under changing climatic conditions [31]. However, our current priority is to characterize the available germplasm for multiple stress tolerant traits for the development of superior germplasm through breeding [9]. Therefore, understanding the complex biological traits underpinning crop yield and stress tolerance is critical [13]. In the present review, we have critically evaluated the information on various individual, combined or sequential stress responses in plants.Wehave also attemptedto identify the common potential molecular components underlying these responses.

## 2. Impact of Individual, Combined and Sequential Stresses on Plants

Despite a considerable increase in the number of abiotic stress related studies conducted during the past decade, most experiments have been focused on the response of plants to individual stress treatment under controlled conditions [24]. In contrast, under field conditions, numerous stresses can occur in combination or simultaneously, and may specifically alter plant metabolism than by individual stress treatments [17]. Due to higher frequency of the concurrent occurrence of multiple stresses under field conditions, the plant response may vary from that tested under laboratory conditions [33]. Therefore, to understand a holistic survival mechanism of plants, it is essential to study the combined and sequential abiotic stresses under the natural environment, which is still far less investigated [34]. Our knowledge of the molecular basis of the additive responses towards combined and sequential abiotic stresses is considerably less [24].

Plants, being sessile organisms, have evolved various physiological and biochemical mechanisms to adapt to extreme environmental conditions during their life cycle [35,36,37]. The level of plasticity against different stresses is regulated by the plant ’s genetic background, along with the duration and severity of stress [15]. Abiotic stress signaling in plants is complex in nature [38] and involves different interacting signal transduction pathways especially during multiple stress tolerance, termed as ‘crosstalk’ [39]. Due to this crosstalk, the outcome of combined or sequential stress can either be neutral, additive, antagonistic, synergistic or sometimes unpredictable in nature (Table 2, Figure 1) [9]. For example, plants increase their transpiration rate by stomatal opening during heat stress, while under combined heat and drought stress, they close their stomata to reduce water loss [40]. Therefore, it is necessary to identify the molecular mechanisms behind the perception and adaptation under combined or sequential stresses [41].

Most of the environmental stresses have similar effects and responses, such as reduction in photosynthesis and growth, hormonal changes, oxidative damage, and the accumulation of stress-related proteins [42]. Besides stomatal closure, root water uptake capacity also plays a significant role in avoiding stress-induced growth reduction during dehydration [43]. Usually, heat stress occurs simultaneously with drought stress under field conditions, which makes studying their combined response indispensable, primarily in drought-stricken and semi-arid regions [44]. Numerous studies have examined the effect of combined heat and drought stress on the development and productivity of maize, barley, sorghum, and different grasses. The co-occurrence of drought and heat stresses would be anticipated largely to alter the physiological and morphological status, and metabolism, especially photosynthesis [45].

Due to intensive irrigation, secondary salinization increases in semi-arid and arid agricultural regions, representing an excellent example of combined drought and salt stress [46,47]. Despite various stress-independent commonalities during osmotic stress, several stress-specific signatures have also been reported in different tissues at the level of transcriptome, metabolome and proteome during individual and combined salinity and drought stresses [11,15]. Under both drought and salinity stress, reduced photosynthesis, improved respiration rate, stomatal closure due to ABA signaling to reduce transpirational water loss and starch breakdown for energy production were observed [15,36].

In contrast to flooding, which restricts root growth, drought stress causes extensive or deeper root systems [48]. However, the duration and magnitude of both drought and flooding might be critical in determining species composition as drought may be lethal due to run-away xylem embolism [49]. Similarly, elevating sea levels and intrusion of seawater causes inland salinization, and along with heavy winds and high temperature, results in salt injury during different phases of the growing season [50,51]. Further, during post-submergence, the limitations on water absorption cause drought-like symptoms, such as leaf wilting, rolling, and decreased relative water content [52]. Irrespective of the common symptoms of low temperature and salinity on plant growth and development [15], limited information is available about their combined effect on plants [53].

**Table 2 ijms-22-06119-t002:** Representative examples showing the specific interactions among various stress combinations on diverse plants.

	Stress Combinations	Crop Plants	Outcomes during Combined Stress	References
**Negative response**	Drought + salinity	Wheat	Reduction in plant growth, biomass and net photosynthetic rate is associated with root length and root K^+^/Na^+^ concentration.	[54]
		Maize	Primary root length significantly reduced under combined stress. 53 metabolites were differentially regulated in primary roots under stress conditions.	[55]
	Drought + high temperature	Tobacco	Stomatal closure, reduced photosynthesis, enhanced respiration, and leaf temperature. Some transcripts induced during individual stress while suppressed during combined stress. Few transcripts were specifically induced during combined stress. Overlap between different transcription factors during individual and combined stress.	[16]
		*Arabidopsis*	454 transcripts were specifically expressed during combined stress. Sucrose, maltose and glucose were highly accumulated under combined stress. Proline only accumulated during drought stress.	[17]
		Wheat	Photosynthesis rate declined under High temperature > Drought > combined stress. High temperature significantly affects grain number, while drought affects grain weight and combined stress affects leaf chlorophyll content, spikelet fertility, total dry weight, and harvest index.	[8]
		*Arabidopsis*	Root allocation increased during drought, while reproductive allocation, hyponasty and specific leaf area increased under high temperature. Origin of accession plays a significant role during individual and combined stress.	[56]
		Maize	Combined stress in comparison to a single stress strongly affected the seminal lateral roots, reducing the dry weight, length, surface area and root mass ratio (RMR).	[57]
	Drought + chilling	Sugarcane	Effect of abiotic stress is cultivar-dependent where the sensitive genotypes were more affected by combined stress than tolerant genotypes. Low root temperature combined with drought severely affects PSII activity.	[58]
	Drought + pathogen	*Arabidopsis*	Interaction among ABA, JA, and ethylene signaling pathways regulate pathogen-, wound-, and dehydration-response and one signaling pathway may dominate over others, depending on the stress conditions.	[59]
		*Arabidopsis*	11 genes differentially regulated, 23 genes specifically regulated, and reduced expression of R-gene mediated response were observed under combined heat, drought, and turnip mosaic virus stress.	[60]
	Drought + UV	Plants	Combined stress induces responses that can be antagonistic, additive or synergistic in comparison to individual stresses which results from interplay between metabolic shuts.	[61]
	Drought + high light	*Arabidopsis*	Mutant seedlings deficient in alternative oxidase (AOX) showed accumulation of anthocyanins in leaves, alterations in photosynthetic efficiency, increased superoxide radical and reduced root growthunder combined stress.	[62]
	Drought + low N	Wheat	Low N stress can lead to accumulation of ABA in wheat seedlings. Combined stress was found to have significant interaction in regulation of genes for nitrate signaling, uptake and assimilation.	[63]
	Drought + heavy metals	Red maple	Combined stress has additive effect in both stems and roots, reductions in hydraulic conductance, xylem-specific conductivity, and leaf-specific conductivity.	[64]
	Drought + soil compaction + mechanical stress	Tobacco	Plant growth and fine root proportion was reduced, while root diameter and xylem area increased under combined stress.	[65]
	Drought + nutrient	Mungbean	Under combined stress, a significant reduction in gas exchange traits (photosynthesis, stomatal conductance, transpiration, instantaneous water use efficiency), and P uptake in seed and shoot were observed under combined stress as compared to individual stress.	[66]
	Salinity +high/low temperature Drought + high/low temperature	Wheat	Root and shoot elongation significantly reduced under individual stress. HT/LT treatment possess additive effect on growth inhibition under salt stress. α-tocopherol significantly increased under drought and salt stress but significantly decreased under HT stress.	[67]
	Salinity + high temperature	*Suaeda salsa*	Combined stress suppressed CO_2_ assimilation and photosystem II efficiency. 57 differentially expressed proteins were observed under individual and combined stress. Expression of nucleoside diphosphate kinase 1, chlorophyll a/b binding protein, and ABC transporter I family member 1 was specifically induced during combined stress.	[68]
	Salinity + pathogen	Rice	Downregulation of *OsMAPK5* expression enhanced pathogenesis-related (*PR*) genes expression and significantly enhanced resistance to fungal (*Magnaporthe grisea*) and bacterial (*Burkholderia glumae*) pathogens but reduced tolerance to drought, salt, and cold. In contrast, overexpression lines exhibited increased OsMAPK5 kinase activity and increased tolerance to drought, salt, and cold stresses.	[69]
	High temperature + ozone	Silver birch (*Betula pendula*)	O_3_ reduces, while temperature increase tree growth and growth may be counteractive during combined stress. R:S ratio decreases under O_3_ exposure. Temperature increase may stimulate soil respiration rates and total biomass, while O_3_ could have opposite effect. Elevated O_3_ decreases C assimilation, foliar C content and productivity.	[70]
	High temperature + pathogen	*Arabidopsis*	NB-LRR type of R or R-like protein is the temperature-sensitive component of plant defense responses. Alterations in the R-like gene *SNC1* and the *R* gene *N* can change temperature sensitivity of defense responses.	[71]
	High temperature + UV-C	Strawberry	Both heat and combined treatments, decreased hue and delayed changes in the colorimetric parameters. The combined stress treatment reduced fungal infections and delayed in vitro germination of *Botrytis cinerea* conidia. Neither the heat nor UV-C irradiation modified the total sugar content, although the combined treatment decreased it slightly relative to the control. The combination of UV-C and heat treatments enhanced the benefits of applying each treatment separately and could be useful to improve and extend strawberry fruit postharvest life.	[72]
	High temperature + high light	Sunflower (*Helianthus annuus*)	Comparative expression analysis of leaves and immature seeds revealed that 89, 113 and 186 genes were differentially expressed in response to HL, HT and HL + HT, respectively.	[73]
	High temperature + CO_2_	Soybean and maize	Increased photosynthetic rates in response to CO_2_ enrichment, while C_4_ cycle is largely unresponsive to increased response to CO_2_ enrichment. CO_2_ enrichment can mitigate the effects of moderately elevated temperatures on crop yield.	[74]
	Low temperature + pathogen	Plants	Both virus and transgene-triggered RNA silencing are inhibited at low temperature. Thus, plants become more susceptible to viruses. RNA silencing-based phenotypes of transgenic plants are lost. However, temperature does not influence the accumulation of micro (mi) RNAs, which play a role in developmental regulation.	[75]
	Low temperature + high light	*Dunaliella salina*	Low temperature and combined high light-low temperature decreased chlorophyll and β-carotene indicating that these treatments cause photo-oxidative stress. High light, low temperature and combined high light-low temperature treatments increased the total ascorbate pool by 10–50% and the total glutathione pool by 20–100% with no consistent effect on their redox state.	[76]
	Pathogen + nutrient	*Arabidopsis*	Field study on potassium disease interaction which provides evidence that facilitated entry and development of pathogens or insects in(to) potassium-deficient plants as a result of physical and metabolic changes is counteracted by an increased defense.	[77]
	UV-B + Heavy metals	Pea	Combined dose (UV-B + 0.01 mM Ni) caused inhibitory effects. Nickel at high doses strongly inhibited PSII activity and the inhibition was further intensified when chloroplasts were simultaneously exposed to UV-B radiation. High doses of Ni (0.1 and 1.0 mM) and UV-B alone interrupted electron flow at the oxygen evolving complex. Similar damaging effects were caused by 0.01 and 0.1 mM Ni together with UV-B, but the damage extended to PSII reaction center, in case of 1.0 mM Ni in combination with UV-B.	[78]
	Nutrient + high CO_2_	*Panicum maximum* Jacq. ‘Mombaça’ (Guinea grass)	Under the combination of [eCO_2_] and elevated temperature [eT] conditions, productivity increases along with the nutritional requirement for N, Ca and S.	[79]
	Heavy metals + heavy metals	Tomato	Zn supply clearly reduced Cd accumulation in leaves and simultaneously increased Zn concentration. Cd-induced oxidative stress in leaves as indicated by an increase in thiobarbituric acid-reactive substances (TBARS) level and chlorophyll breakdown. Zn supplementation, at low level, restored and enhanced the functional activity of these enzymes (SOD, CAT, APX and GR) as compared to Cd-alone-treated plants. The beneficial effect of adequate Zn level on Cd toxicity was confirmed by a significant decrease in TBARS level and restoration of chlorophyll content. However, when Zn was added at a high level in combination with Cd, there was an accumulation of oxidative stress, which was higher than that for Cd or excess Zn alone treatments.	[80]
Positive response	Drought + ozone	Birch	Combined stress increases the N concentration in the leaves, the thickness of the upper epidermal cell wall, the number of pectinaceous projections of mesophyll cell walls, and the vacuolar tannin-like depositions and phenolic droplets, which are regarded as signs of activated stress defense mechanisms. The increase in specific foliage mass, cytoplasmic lipids (younger leaves), and a condensed appearance of the upper epidermal mucilaginous layer were caused by both drought and ozone but were not additive.	[81]
		beech trees (*Fagus sylvatica*)	Photosynthesis (*A*max), stomatal conductance (*g*s), and electron transport rate (ETR) were lowered during drought rather than ozone, whereas chlorophyll levels did not differ. Comparison of AOT40 [Accumulated Ozone exposure over a threshold of 40 ppb ((80 µg/m³)], an O_3_ exposure-based risk index of O_3_ stress, and cumulative ozone uptake (COU) yielded a linear relationship throughout humid growth conditions. The findings support the hypothesis that drought protects plants from O_3_ injury by stomatal closure, which restricts O_3_ influx into leaves and decouples COU from high external ozone levels. High AOT40 erroneously suggested high O_3_risk under drought. Enhanced ozone levels did not aggravate drought effects in leaves and stem.	[82]
		*Medicago truncatula*	*Medicago truncatula* cultivar Jemalong that is sensitive to ozone and drought stress when applied singly, showed tolerance when subjected to a combined application of these stresses. Lowered stomatal conductance may be a vital tolerance mechanism to overcome combined ozone and drought. Sustained increases in both reduced ascorbate and glutathione in response to combined stress may play a role in lowering reactive oxygen species and nitric oxide toxicity. Transcriptome analysis indicated that genes associated with glucan metabolism, responses to temperature and light signaling may play a role in dampening ozone responses due to drought-induced stomatal closure during combined occurrence of these two stresses. Gene ontologies for jasmonic acid signaling and innate immunity were enriched among the 300 differentially expressed genes unique to combined stress. Differential expression of transcription factors associated with redox, defense signaling, jasmonate responses and chromatin modifications may be important for evoking novel gene networks during combined occurrence of drought and ozone. The alterations in redox milieu and distinct transcriptome changes in response to combined stress could aid in tweaking the metabolome and proteome to annul the detrimental effects of ozone and drought in Jemalong.	[83]
	Drought + high CO_2_	Plants	Elevated atmospheric CO_2_ cause an increase in leaf and canopy photosynthesis, especially in C3 plants, with minor changes in dark respiration. Additional CO_2_causes an increase in biomass without marked alteration in dry matter partitioning, reduced transpiration of most plants and improvement in WUE. However, spatiotemporal variation in these attributes impact agronomic performance and crop water use in a site-specific manner. Nutrient acquisition is closely associated with overall biomass and strongly influenced by root surface area.	[84]
	salinity + High temperature	tomato	The combination of heat and salinity provides a significant level of protection to tomato plants from the effects of salinity. We observed a specific response of plants to the stress combination, which included accumulation of glycine betaine and trehalose. The accumulation of these compounds under the stress combination was linked to the maintenance of a high K^+^concentration and thus a lower Na^+^/K^+^ratio, with a better performance of the cell water status and photosynthesis as compared with salinity alone.	[14]
	Salinity + hypoxia	*Salix*	Combined stress favored root biomass production increasing number and elongation of roots.	[85]
	Salinity + high CO_2_	lettuce	Elevated CO_2_ and its combination with salinity or high light increases biomass production. Elevated CO_2_ and its combination with salinity or high light increases the antioxidant capacity, while high light treatment alone increased the antioxidant capacity of red-leaf lettuce, but not of green-leaf lettuce.	[86]
	Salinity + boron	*Zea mays*	Under salt stress, the activity of specific membrane components can be influenced directly by boron, regulating the water uptake and water transport through the functions of certain aquaporin isoforms.	[87]
	Ozone + pathogen	Plants	Cellular responses to these environmental challenges are rather similar, which might be the reason why plants that are resistant to one stress are sometimes cross-tolerant to others.	[88]
		Microbes	The acetate, propionate, and butyrate buffered aqueous ozone combinations had a significant 3–4 log reduction of *S. aureus* (*p* < 0.05) colony forming unit (CFU), while citrate or oxalate buffered aqueous ozone, statistically significant versus buffer alone, had less activity.	[89]
	Ozone + UV	*Escherichia coli*	Ozone was found to be a stronger disinfectant than UV radiation, using both simultaneously was more effective than using them individually.	[90]
	Ozone + high CO_2_	Rice	Elevated CO_2_ (627ppm) increases rice yields by 23%. Modest increases in grain mass and larger increases in panicle and grain number contributed to this response. The response of rice to elevated CO_2_ varied with fumigation technique. The more closely the fumigation conditions mimicked field conditions, the smaller was the stimulation of yield by elevated CO_2_. Free air concentration enrichment (FACE) experiments showed only a 12% increase in rice yield. When compared with rice grown in charcoal-filtered air, rice exposed to 62° ppb O_3_ showed a 14% decrease in yield. Many determinants of yield, including photosynthesis, biomass, leaf area index, grain number and grain mass, were reduced by elevated O_3_.	[91]
	Pathogen + UV	Various plants	Cellular responses to these environmental challenges are rather similar, which might be the reason why plants resistant to one stress are sometimes cross-tolerant to others.	[88]
	High CO_2_ + high light	lettuce	High light treatment alone increased production in green-leaf lettuce but not in red-leaf lettuce. On the other hand, elevated CO_2_and its combination with salinity or high light increased the antioxidant capacity, while high light treatment alone increased the antioxidant capacity of red-leaf lettuce, but not of green-leaf lettuce.	[86]

### 2.1. Physiological, Growth and Developmental Processes

Plants induce various interacting signal transduction pathways when exposed to different stress combinations [39]. In general, combined or sequential stress evoke distinct responses in plants than individual stresses in physiological, molecular and metabolic networks, which influence nutrient assimilation and distribution [32], yet share common pathways and responses. Studies applying combined stress scenarios to mimic field conditions are increasing [11]. These studies strongly focus on comparative transcriptomics of abiotic and biotic interactions. The integration of various metabolic pathways and the crosstalk between different sensors and signal transduction pathways further augments combined stress response [92]. Current studies on the transcriptome analysis of combined stress response mainly represent a snapshot of a single time point. The sequential stress exposures induce priming [93], which allows plant to respond rapidly in future environmental vagaries [11]. Combined stress results in oxidative stresses, which modulate sugar levels, plant growth and stress responses [11]. This has allowed the characterization of genes specific to individual, combined or sequential stress conditions [11].

As recent research has indicated, temperatures higher than 35 °C affect germination, vegetative, reproductive, grain filling stages and ultimately yields [94]. However, the reproductive stage is more sensitive to combined drought and heat stress, whilst each stress differentially affects reproductive traits [7]. Earlier reports demonstrated that combined drought and heat stress shows similar tolerance mechanism to individual stresses, including the accumulation of compatible solutes, protective proteins, activation of non-enzymatic and enzymatic antioxidant system [95,96]. Similarly, alterations in physiological processes, including photosynthesis, lipid accumulation, oxidative metabolism, and transcript abundance was observed [10], affecting membrane stability, stomatal conductance, reduced leaf area and water-use efficiency [97,98].

Under field conditions, heat and drought stresses occur simultaneously resulting in strikingly varied responses that cannot be inferred from their individual responses [99], which is also species specific [40]. High temperature causes stomatal opening to increase transpiration and leaf surface cooling, long, slender leaves with a higher specific leaf area and decreased root development [60]. In contrast, drought reduces leaf stomatal conductance and leaf area, resulting in enhanced canopy temperature by 2–5°C, while improved root development to prevent water loss and ABA accumulation [36,100]. Whereas, stomata remain closed during combined drought and heat stress, leading to reduced membrane stability, relative water content, plant length, shoot fresh/ dry weight, stem diameter, leaf area, kernels/ear, 100-kernel weight, harvest index, seed abortion and reduced yield in potato, wheat and maize [101,102,103]. In addition, enhanced respiration rate due to breakdown of reserved assimilate provides energy for acclimation under combined drought and heat stress to mitigate CO_2_ assimilation loss [104]. Therefore, under natural field conditions, co-occurrence of heat and drought stress requires different strategies for acclimation [92].

Global climate change involvesincreased flooding events, which is detrimental to plant growth and productivity in agricultural ecosystems [105]. Different parameters, such as leaf area, shoot dry weight, photosynthesis, transpiration, absorption and transport of nutrients, panicle number, panicle weight, harvest index drastically reduce under turbid floodwater [51,106]. To confer enhanced adaptation and survival during energy starvation, plants develop mechanisms to survive during transient influx of water, which include energy generation through fermentation under hypoxia, adventitious roots/aerenchyma development for aeration, petiole and internode elongation to outgrow submergence, reduction in epidermal cell wall and cuticle thickness for reduced diffusion resistance [106,107]. In addition, gas films on the leaf surface hamper the salt entry into leaves [106]. Further, after desubmergence plants were abruptly exposed to higher oxygen and light intensity causing oxidative damage to photosystem II reaction centers and desiccation of leaves due to reduced hydraulic conductivity in shoots and mineral leaching [108]. Coastal flooding causes combined salinity and submergence, causing oxygen deprivation and restricted energy production for ion transport resulting in accumulation of Na^+^ and Cl^-^ and reduces K^+^ concentration. Halophytes inhabiting coastal regions preserve more pigment, chloroplast structure [109], develop aerenchyma and maintain antioxidant systems and activate the hormonal and signaling pathways [106].

### 2.2. Photosynthesis and Respiration

Photosynthesis, i.e., photo-assimilate production and carbon assimilation are the most sensitive physiological processes to adverse environmental conditions [110]. Improved integrity of the photosynthetic apparatus often holds the key for stress tolerant genotypes. Photosystem II electron transport is the most sensitive segment of the photosynthetic machinery [15] and its structural and functional ability gets disrupted under adverse environments [15]. Drought stress regulates photosynthesis through stomatal closure and reduced CO_2_ uptake and diffusion into mesophyll tissues, favoring oxygenase activity (Figure 2) [111,112], decline in ribulose 1,5-bisphosphate carboxylase/oxygenase (Rubisco) and ribulose bisphosphate (RuBP) activity [112], impairment of ATP synthesis, and photo-phosphorylation resulting in decline in crop biomass and yield [113]. Similarly, at temperatures higher than 35 °C, photosynthesis becomes considerably reduced [114]. At higher temperatures, oxygen solubility and Rubisco activity is reduced, causing higher photorespiration and lower photosynthesis [115]. In addition, high temperature enhances thylakoid membrane fluidity, leading to dislodging of PSII light harvesting complexes from thylakoid membrane, indicated by steep rise in basal level of chlorophyll fluorescence [116]. Similarly, cold and salt stress individually render adverse effects on photosynthetic electron transport chain by impairing performance of photosynthetic rate and photochemical efficiency in crops such as sunflower, bean and maize [53,117,118,119]. However, limited information related to the effects of combined salt and cold on photosynthesis is available [53]. Similarly, salinity and drought also negatively affect photosynthesis by limiting internal CO_2_ through stomatal closure [11].

Mitochondrial respiration plays a pivotal role in determining the growth and survival of plants and is reduced under drought and temperature stress [120]. However, under field conditions, high temperature stress is associated with soil temperature and drought. Increased respiratory losses by grains or kernels under heat stress can offset the increased influx of assimilation resulting in higher yield losses [121]. Both stresses (heat and drought) increases membrane fluidity, leakiness and reduces integrity of the proteins and membranes. This leads to a decline in photosynthesis before respiration losses and enhanced photorespiration. Therefore, both drought and heat stress combination may, thus, be additive or multiplicative and exacerbates each other effects [122,123]. Combined drought and heat stress influences diurnal, as well as seasonal patterns of leaf water potential, carbohydrate content, photo-assimilate translocation and stomatal conductance associated with senescence [94,124]. The response pattern of crops under heat and drought stress during various growth stages can be the basis for selecting multiple stress tolerant variety to solve yield stability and nutritional crisis [113]. Similarly, flooding stress reduces hydraulic conductance of roots [125], caused either by oxygen deprivation or accumulation of CO_2_ around the roots [126]. This involves signal transduction from the hypoxic/ anoxic root system to the shoot, and subsequently its perception and conversion into physiological responses, such as drastically reducing energy production through eliminated/ reduced mitochondrial respiration. The energy requirements for survival are produced through fermentation pathways, primarily ethanolic fermentation [127,128].

### 2.3. Reactive Oxygen Species (ROS) Homeostasis

During suboptimal environmental conditions, different pathways are affected differently, which disrupts cellular homeostasis accompanied by the production of reactive oxygen intermediates (ROIs) due to increased electron flow from disrupted pathways to oxygen reduction [129,130]. Different studies demonstrated that combined and sequential stresses trigger various ion channels leading to hormonal changes, which in turn, generated unique set of reactive oxygen species (ROS), i.e., O_2_^−^ and H_2_O_2_ that causes cellular damages termed as ‘oxidative stress’ (Figure 3) [131,132]. ROS are produced in almost every cellular organelle, primarily having high oxidized metabolic activity (chloroplasts) or high electron flow rates, during numerous enzymatic reactions and are important signaling molecules within the cell and cellular communication in between different cells [41]. Cellular ROS levels are regulated by antioxidant molecules, i.e., carotenoids, tocopherols, alkaloids, phenolic compounds, flavonols, GSH, ascorbate and enzymes i.e., AOX, CAT, APX, SOD, GPX, glutathione reductase, glutathione-S-transferase and peroxidases that keep ROS levels in balance and protects against redox-regulated defense [110,129,130,133,134]. It was shown earlier that cytosolic APX1(*apx1*) or ABA function deficient mutants and ROS-regulated protein PP2Cs (*abi-1*) are sensitive to combined heat and drought stress response [135,136,137].

Previous research has revealed a regulatory network of submergence-induced signal transduction through hormonal regulators, ROS, and ethylene regulating metabolic responses besides morphological adaptations for survival [52]. Post-submergence stress is caused by sudden reoxygenation after prolonged hypoxia/ anoxia and re-illumination, after acclimation to low light under water. This causes electron leakage in membranes and electron-transport chains leading to inactivation of photosystem reaction centers and burst of ROS production [138]. Less ROS-accumulating genotypes display better recovery after de-submergence [52]. The production of ROS singlet oxygen (^1^O_2_) under excessive light inhibits D1 protein synthesis and hampers the repair of photodamaged PSII, dampened carbohydrate replenishment and senescence. Similarly, under heat stress, electrons from NADH produced by the soluble, membrane-bound complexes at the inner mitochondrial membrane are disrupted or uncoupled [139,140]. During stress acclimation, plants undergo metabolite changes that involve reactive oxygen intermediates (ROI) production and defense regulated genes during stress, such as ROI scavenging enzymes and heat shock proteins [141]. Various studies have demonstrated that plants display higher ROS burst and antioxidant enzyme activity under individual and combined cold and salt stress [130] or drought and salinity [142], such as rice [140], Azolla [143], wheat [144], and barrel clover [145]. Previous studies have also shown dehydration and submergence tolerance through ROS detoxification regulatory enzyme and their transcriptional regulation in tobacco, rice, and *Arabidopsis* [108,139].

### 2.4. Multi-Omics Approach: New Potential Key Mechanisms

Significant developments have been made in plant genomics, particularly contributing to the development of stress tolerant crops. Transcript profiling under individual and combined stresses showed significant differences in plants [133,146], indicating differential regulation of combined and sequential stresses under individual and combined stresses [7]. Yet, little is known about the overlapping stress combinations and their genetic interplay of unique-differentially expressed genes (DEGs) during plant acclimations the well-documented phenomena of “cross tolerance” [32]. Therefore, the major challenge is to develop multiple stress tolerant cultivars with lack of information on physiological and molecular mechanisms. The complex signaling pathways associated with stress sensing and activation of defense and acclimation pathways involve mitogen-activated protein kinase cascades, calcium-regulated proteins, ROI, and cross talk among various transcription factors [39,147,148]. Interestingly, different stress conditions can trigger similar stress response pathways [39,149]. Various studies have been conducted to delineate the cellular metabolism of crop species, but despite recent advancements in genomics, metabolic adaptations under multiple stresses remain poorly characterized and require a systematic understanding ofhighthroughput “omics” combined within silico metabolic modeling [128].

#### 2.4.1. Transcriptomics

Despite the continuous generation of transcriptomic data from various studies, our understanding of plant responses against combined or sequential stress is incomplete. Comparative transcriptomic studies have revealed molecular cross talks due to differential accumulation of both “unique genes” or “shared genes” during individual or combined or sequential stress (Figure 4) [150,151]. Shared genes between individual and combined stress have mostly demonstrated the differential expression of transcription factor (TF)-encoding genes, polyamine and primary carbon metabolism related genes and phytohormone pathway-related genes [152]. Comparative transcriptomics in *Arabidopsis* plants treated with six abiotic stresses (osmotic, oxidative, salinity, drought, heat and cold) and one biotic stress (*Botrytis cinerea*) revealed upregulation of 3 genes and downregulation of 12 genes under individual stresses and 13 genes commonly upregulated under heat/salinity/osmotic stress/*B. cinerea* stress, while 29 genes were commonly down-regulated [153]. Previous workers have demonstrated that members of MAPK family were differentially regulated during various abiotic and biotic stresses and differentially regulate their downstream gene expression and signaling responses [41]. MAPK pathways also show cross talk with ABA signaling pathways, ROS and ethylene [154,155]. In previous studies, the altered expression of various micro-RNAs and their gene regulation in transgenic plants have been observed during various stress experiments [11,156]. In addition, Ca^2+^ ions also play a key role during abiotic stress signaling, where they enter the cell through Ca^2+^-permeable channels to regulate specific downstream responses, such as protein interactions and stress-responsive gene expression [157].

Previous studies on heat and drought stress revealed altered physiological changes that are reflected in their gene expression patterns subjected to combined stress [17]. In wheat, large overlaps were reported in commonly up-regulated or down-regulated genes during individual and combined stress [158]. Similarly, in other plants, the unique transcript profile was observed in response to combined heat and drought stress acclimation [159]. Further, rapid changes in various transcription factor families (DREB, NAC, MYB, bZIP) was observed, which further activated downstream stress response pathways and were coordinated through synergistic or antagonistic interactions of metabolic and hormonal pathways under different abiotic stresses [160].

To improve the productivity of rice in coastal areas, genotypes tolerant to both water stagnation and salinity stress together can improve breeding efficiency [50]. It was reported earlier that submergence tolerance gene SUB1A imparts tolerance only under the vegetative stage, neither in germination nor at the reproductive phases [93]. A polygenic locus encoding two APETALA2/ Ethylene Response Factor (AP2/ERF) DNA binding proteins, SNORKEL1 and SNORKEL 2 (SK1 and SK2) provides an escape response by downregulating brassinosteroid synthesis and promoting gibberellic acid regulated internode elongation in deep-water rice [106,161]. In contrast, submergence1A-1 (Sub1A-1) restricts shoot elongation for adaptation and induces alcohol dehydrogenase genes for carbohydrate production [52,162]. A traditional rice cultivar “Baliadhan,” with SUB1 QTL is flooding tolerant during reproductive stage more than Swarna and Swarna-Sub1 [51,163]. However, cultivar FR13A (Dhalaputia) tolerates flooding even more easily because of long stature [164] and retaining airy zones (gas films) around their leaves for longer durations [51]. These gas films help in maintaining the carbohydrate status and internal aeration in rice during submergence. In contrast, FR13A is susceptible to salinity, but showed similar response during submergence and salinity in coastal areas [165]. It shows that submergence tolerance is independent of submergence inducible genes SUB1B and SUB1C, located at SUB1 locus in rice [51]. SUB1A was also reported to delay chlorophyll and carbohydrate breakdown in aerial tissue under prolonged darkness [106,166] and the loss of function mutants (*prt6-1* and *ged1*) showed starch accumulation in leaves during submergence, providing enhanced survival [167].

During sequential submergence and post-submergence stress, ethylene is produced rapidly and accumulates in the cell membrane. Here, it binds with ethylene receptors and stabilizes ETHYLENE-INSENSITIVE3-LIKE1 (EIL1) and ETHYLENE-INSENSITIVE3 (EIN3) transcription factors and regulates downstream genes leading to a sequential stress response, including leaf hyponasty, shoot elongation, and adventitious root formation [106,168]. SUB1A, avoids unnecessary energy consumption due to gibberellin (GA)-mediated elongation of submerged tissues [37,169,170]. SUB1A also increases brassinosteroids (BR) production, which enhances SLENDER RICE1 (SLR1, a DELLA protein) accumulation and further bioactive GA degradation [171]. OsETOL1 was found to negatively regulate ACC and ethylene production under drought and inhibits carbohydrate transportation to the developing seeds from leaves, leading to reduced grain filling and spikelet fertility. In contrast, OsETOL1 promotes carbohydrate consumption and energy production during submergence causing leaf elongation [37]. Similarly, sucrose-nonfermenting1-related protein kinase1A (SnRK1A), an ortholog of mammalian adenosine monophosphate-activated protein kinase (AMPK) and yeast sucrose nonfermenting1 (SNF1) is reported as a carbohydrate starvation/energy depletion sensor during submergence. Further, trehalose-6-phosphate (T6P) inhibits SnRK1 activity and conversion of T6p into trehalose-6-phosphate phosphatase (TPP7, located at submergence tolerance QTL, qAG-9-2), increases sink strength in coleoptiles and germinating embryos, thereby, improving starch mobilization and seed germination under submergence [106,172,173].

SUB1A was also reported to act as convergence point during sequential drought after de-submergence in rice through detoxification of ROS, and stress-inducible gene expression and prevents reduction of hydraulic conductivity in leaves after de-submergence [108]. Microarray and qPCR studies have shown that ethylene biosynthesis and signaling genes gets suppressed under drought stress and overexpression of homeobox (HB) genes regulated by different clades of HD-Zip type transcription factor, enhanced tolerance against various abiotic stresses [174]. Numerous transcription factors were identified having functional roles in transcriptional regulation under drought stress such as DREB/CBF, ABRE, AREB/ABF, NAC and ERF [175,176]. SUB1A induces accumulation of these transcription factors along with SLR1 and SLRL1 in ABA-dependent or ABA-independent manner for providing dehydration tolerance [108].

Salinity and drought alters the ionic and osmotic signal pathways respectively in different crops. Various QTLs and transcription factors have been characterized for salt stress (osmotic and ionic) tolerance during different developmental stages such as *NAC* [177], *PDH45* [178], *Saltol* [51], *Hardy* [179], *HKT* [180], *NHX* [181,182] in rice and *SOS* [183] in *Brassica*. *SOS* signaling pathway is mostly explored during osmotic stress signaling [184,185], since identification of first two-component phosphorelay system (TCS) osmosensor in plant *AHK1* [186]. Heterologous expression of various genes in rice showed better ROS detoxification mechanism and reduce membrane damage during salinity and drought stress such as *OsMT* [187], *OsCPK9* [188], *MDCP* [189], *CrRLKs* [41] and *TPSP* [140]. Similarly, Open Stomata1 (*OST1*), a member of SNF1-related protein kinases2 (SnRK2) protein kinase family is a central regulator of cold signaling pathway [190] besides ABA dependent stomatal closure during osmotic stress [191]. *OST1* phosphorylates *BTF3* (Basic Transcription Factor3) and *ICE1* (Inducer of CBF expression1) transcription factors, leading to the expression of *COR* (Cold-Regulated) genes [192]. Similar studies have demonstrated that unsaturated triacylglycerols accumulation through Phospholipid: Diacylglycerol Acyltransferase1 (*PDAT1*) increases membrane fluidity during high temperature [193].

#### 2.4.2. Proteomics

Abiotic stress has profound impacts on plant proteomes, which include alterations in their localization, post-transcriptional and post-translational modifications, molecular cross-talks and biochemical interactions. Therefore, proteins shape novel phenotype and altered biological traits under environmental vagaries, which include transcriptional regulation of osmotins, dehydrins, LEA, and NAC transcription factors [194,195,196]. Various functions of proteins have been reported under individual stress while their response under combined stress is still very less. Different reports have confirmed upregulation of enzyme pathways leading to enhanced accumulation of osmoprotectants under salinity and drought stress [197]. The signaling pathways related to differentially expressed proteins during individual and combined stress responses were reported to be different from each other. Of these, some exclusively responded during individual and combined stress, while few showed similar function during individual and combined stress, such as protein kinases, phosphatases, LEA, dehydrins, osmotins, and HSPs [19]. Osmotic (drought and salinity) stress sensory proteins are mostly localized in the plasma membrane and demonstrate similar response individually but differ during combined stress [40,198]. Similar reports were observed during proteomics and western blot analysis in response to combined drought and heat stresses as glycolate oxidases, catalases and dehydrins were up-regulated during drought, while thioredoxin peroxidase is up-regulated during heat stress. However, during combined stress, ascorbate peroxidases were specifically down-regulated, while HSPs, alternative oxidase, glutathione peroxidase, cyclophilin, WRKY, phenylalanine ammonia lyase, and ethylene-responsive element-binding protein were upregulated [199,200,201].

#### 2.4.3. Metabolomics

Climate change has exacerbated the unpredictability and severity of environmental vagaries that are sub-optimal for plant growth and survival. Plant resilience to environmental extremes continuously adjust plant metabolism to regulate growth and development within a highly dynamic, and often, inhospitable environment, as well as after the removal of stress. Numerous inter-connected signaling pathways that regulate metabolic networks revealed differential regulation of physiological and biochemical production of secondary metabolites during abiotic stresses, as well as contribute markedly to antioxidant defense response [202,203]. Unfortunately, previous studies on plant metabolites have been focused on individual stresses [204,205], which were later found to differ during combined and sequential stress response [47]. However, few studies on individual and combined abiotic stress reported some uniquely accumulated metabolites during individual stress such as proline, while few compounds accumulated under combined stress, such as sucrose, maltose, and glucose [17]. Therefore, metabolic plasticity may activate appropriate defense responses to cope with multiple environmental stresses [206]. The metabolite profiling of samples during individual and combined abiotic stresses led to the identification of metabolic markers which were closely related [24].

It was hypothesized that secondary metabolite accumulation and related gene expression during drought and salt stress may stimulate osmotic adjustment [207]. Different stress-inducible metabolic networks and signal transduction pathways are triggered during early and later stages of stress to achieve global metabolic homeostasis [208,209]. This reveals greater crosstalk between the metabolic pathways of individual or combined abiotic stresses during different developmental stages [24,210,211].

## 3. Conclusions and Prospective

Current studies on plant stress response have mainly been focused on individual stress conducted under laboratory conditions. However, to gain a meaningful understanding of the actual field conditions, combined and sequential stress responses need to be thoroughly studied. Previous studies have revealed that combined and sequential stresses act differently or similarly and evoke distinct networks than their individual counterparts. All individual, combined and sequential stresses changes phytohormone balance and nutrient assimilation pattern leading to oxidative stress, as well asreduced growth and yield. Taken together, the distinctive involvement of various stress-responsive transcripts, proteins and metabolites during individual, combined and sequential stresses, suggest unique cellular defense responses. However, detailed analysis of pathways and associated genes during individual, combined and sequential stresses are largely unpredictable. Emerging information about signal integration and stress-signaling pathways can provide information on gene functions to develop advance breeding programs for tailoring stress tolerant genotypes. Despite major advances in elucidating abiotic sensing mechanisms, the identification of bonafide sensing mechanisms during individual, combined and sequential abiotic stresses will be a boon in delineating cellular signaling pathways and their responses during complex environmental conditions.

## Figures and Tables

**Figure 1 ijms-22-06119-f001:**
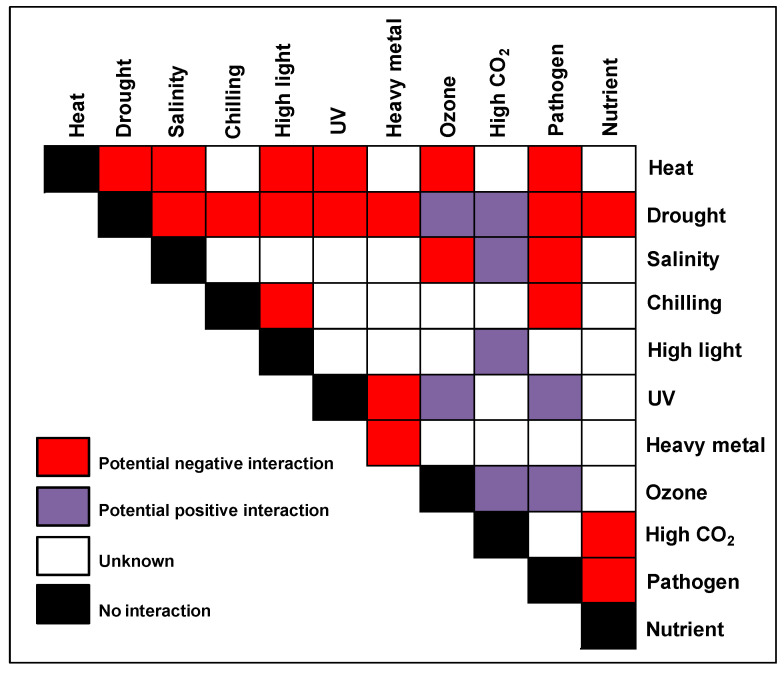
The stress matrix. Different combinations of potential environmental stresses that can affect crops in the field are shown in the form of a matrix. The color of the matrix indicates stress combinations that were studied with a range of crops and their overall effect on plant growth and yield. References for the combined studies are given in the Table 2.

**Figure 2 ijms-22-06119-f002:**
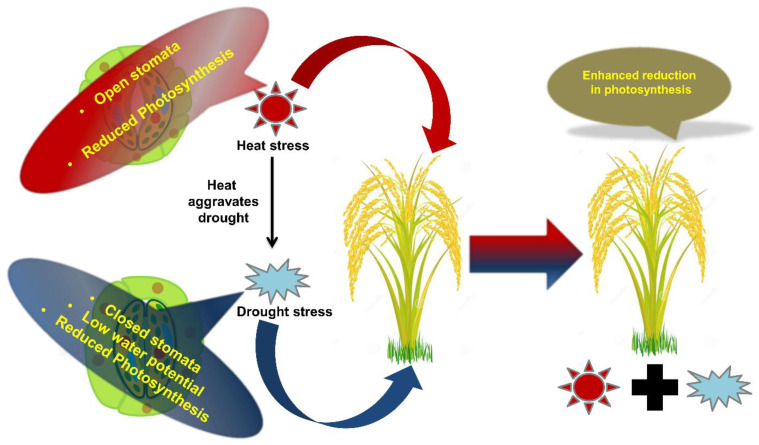
Schematic representation showing the effects of combined stress on plants. Effect of combined stresses on plants is explained taking an example of heat and drought. For example, simultaneous exposure to heat and drought leads to enhanced retardation of physiological processes such as photosynthesis.

**Figure 3 ijms-22-06119-f003:**
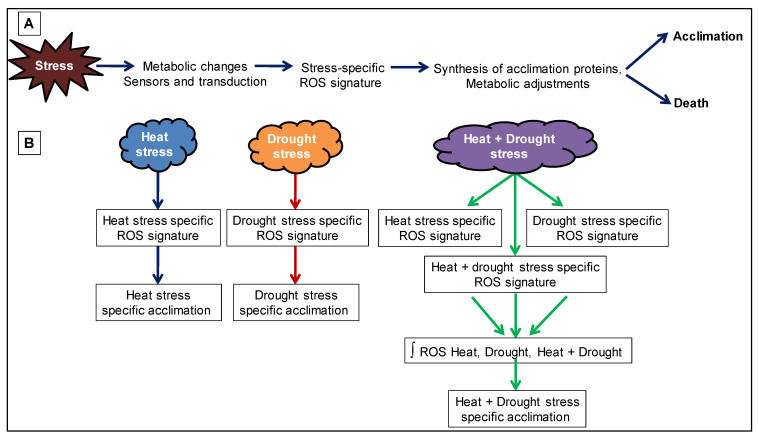
Reactive oxygen species (ROS) signature during abiotic stress combination. (**A**) Abiotic stress is shown to result in the formation of a ROS signature that mediates plant acclimation or cell death. (**B**) A combination of two different stresses (heat and drought stress) is shown to generate a ROS signature that is unique to the stress combination and is the result of combining three different ROS signature (ROS signature for heat stress, ROS signature for drought stress and ROS signature generated from the combination of heat + drought stress).

**Figure 4 ijms-22-06119-f004:**
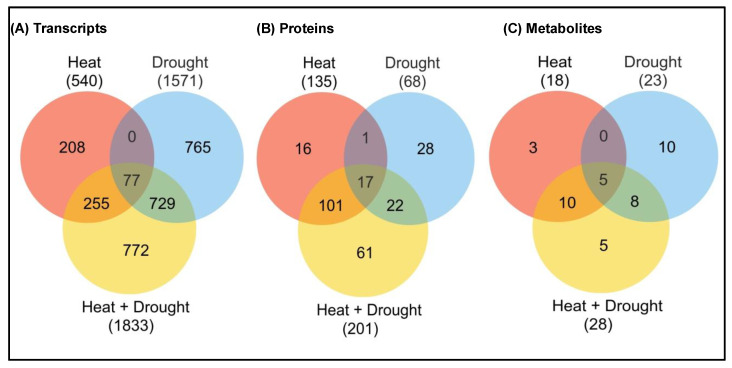
Unique characteristics of multi-omics studies under heat, drought and combined heat + drought stress. Venn diagram showing the overlap between (**A**) transcripts, (**B**) proteins, (**C**) metabolites (up-regulated or down-regulated) during heat or drought stress, or a combination of heat and drought stress. The total number of transcripts or metabolites is indicated in parenthesis. The stress-induced expression was based on a significant linear regression (*p* < 0.01) and a threshold of ≥1.5-fold (log_2_) over control [17,19].

**Table 1 ijms-22-06119-t001:** Representative examples showing the physiological and molecular processes studied in plants in response to combined high temperature and drought stress (HT+D).

S.No.	Processes Studied	Crops	References
1	Gene expression	Tobacco	[16]
2	Transcriptome analysis	*Arabidopsis*	[17]
3	Morpho-physiological traits	Agricultural crops	[8]
4	Morpho-physiological traits	Agricultural crops	[11]
5	Reactive Oxygen Species (ROS)	Agricultural crops	[18]
6	Physiological and Proteome changes	Maize	[19]
7	Proteome changes	Agricultural crops	[20]
8	Proteome changes	Rice	[21]
9	Anti-oxidative enzymes, ABA response and Proteome changes	Maize	[22]
10	Physiological and gene expression response	*Camellia oleifera*	[23]
11	Metabolic response	Maize	[24]
12	Metabolic response	Rice	[25]
13	Grain yield	Sorghum	[26]
14	Grain growth and starch accumulation	Barley	[27]
15	Genetic studies	Maize	[28]
16	Antioxidant metabolism and lipid Peroxidation	Turfgrasses	[29]
17	Physiological recovery	Kentucky bluegrass	[30]

## Data Availability

All data and materials are available upon reasonablerequest from the corresponding author.

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
