# Peer review of "Elucidating the Response of Crop Plants towards Individual, Combined and Sequentially Occurring Abiotic Stresses"

_ijms, 2021, doi:10.3390/ijms22116119_

Round 1

Reviewer 1 Report

Khalid Anwar et al. comprehensively summarized the response of crops plants under various stress including their transcriptomics, proteomics and metabolomics.

Major points

Title of the article might be revised to - Elucidating the potential commonalities and conflicts in the response of crop plants towards individual, combined and se-quentially occurring abiotic stresses

Include more information in Table 2 from various crops including response of both roots and shoots of plants under various abiotic stresses.

Line 130-131: Response of plants roots under drought and salinity differs. Revise and write comprehensively

Revise reference section extensively. Italicize all botanical names.

Minor points

Line 37, 87: “burgeoning’ – use more commonly used words for the international audience.

Table 2, point 16: Antioxidant

Line 76: reference [9,32], similarly line 427, 435

Line 88-89: revise grammar

Line 109: that induces

Italicize ‘Arabidopsis’ throughout

Reference 75, Table 2: Do you mean heavy metals interactions?

Use H2O2, CO2 throughout

Line 384: high throughput

Line 396, 398: Botrytis cinerea italicize

Line 405, 406: Ca2+

Line 458: [99, 165, 166]

Reviewer 2 Report

This manuscript summarizes information about stress responses in plants under combinational abiotic stress . It is well documented in this manuscript. I listed some minor points I noticed below. I hope it will be helpful to improve this manuscript.  

Minor points

Line 317 Please put apx1 and abi-1 in italic.

Line 353 Add a parenthesis at the end of the sentence.

Line 436 Please put prt6-1 and ged1 in italic.

Line 458 Add a comma between 165 and 166.

Line 470-485 I recommend authors clarify which plant species in the story here. Arabidopsis, rice, or another?

Figure 4 What is the cutoff for these data here? Please describe the information like two times fold difference compared to control etc. Besides, please explain more details about proteins and metabolites and what kind of relationship authors found among these data if the authors show these three data, but not only transcripts.  

Table1 Some words sound similar in this table. For example, Proteome changes and proteomic response; Metabolite profiling and Metabolic response. Is there any significant difference between these words? Please provide a little more information about the difference.
